# Colistin Treatment Affects Lipid Composition of *Acinetobacter baumannii*

**DOI:** 10.3390/antibiotics10050528

**Published:** 2021-05-03

**Authors:** Ye Tao, Sébastien Acket, Emma Beaumont, Henri Galez, Luminita Duma, Yannick Rossez

**Affiliations:** Enzyme and Cell Engineering UPJV, UMR CNRS 7025, Centre de Recherche Royallieu, Université de Technologie de Compiègne, CEDEX, 60205 Compiègne, France; ye.tao@utc.fr (Y.T.); sebastien.acket@utc.fr (S.A.); emma.beaumont@orange.fr (E.B.); henri.galez@free.fr (H.G.); luminita.duma@utc.fr (L.D.)

**Keywords:** colistin resistance, untargeted lipidomics, *Acinetobacter baumannii* isolate, glycerolipids, fatty acid content

## Abstract

Multidrug-resistant *Acinetobacter baumannii* (*A. baumannii*) causes severe and often fatal healthcare-associated infections due partly to antibiotic resistance. There are no studies on *A. baumannii* lipidomics of susceptible and resistant strains grown at lethal and sublethal concentrations. Therefore, we analyzed the impact of colistin resistance on glycerolipids’ content by using untargeted lipidomics on clinical isolate. Nine lipid sub-classes were annotated, including phosphatidylcholine, rarely detected in the bacterial membrane among 130 different lipid species. The other lipid sub-classes detected are phosphatidylethanolamine (PE), phosphatidylglycerol (PG), lysophosphatidylethanolamine, hemibismonoacylglycerophosphate, cardiolipin, monolysocardiolipin, diacylglycerol, and triacylglycerol. Under lethal and sublethal concentrations of colistin, significant reduction of PE was observed on the resistant and susceptible strain, respectively. Palmitic acid percentage was higher at colistin at low concentration but only for the susceptible strain. When looking at individual lipid species, the most abundant PE and PG species (PE 34:1 and PG 34:1) are significantly upregulated when the susceptible and the resistant strains are cultivated with colistin. This is, to date, the most exhaustive lipidomics data compilation of *A. baumannii* cultivated in the presence of colistin. This work is highlighting the plasma membrane plasticity used by this gram-negative bacterium to survive colistin treatment.

## 1. Introduction

*Acinetobacter baumannii* (*A. baumannii*) is a member of the ESKAPE group, which is the main bacterial group that causes infections in humans [1]. This emerging pathogen is a major cause of hospital-acquired infections frequently associated with bloodstream infection and pneumonia [2]. *A. baumannii* has been ranked as a bacterium that poses the greatest health threat by the World Health Organization and for which new antibiotics are desperately needed [3]. As a gram-negative bacterium, the *A. baumannii* outer membrane consists of a monolayer of glycerophospholipids and an exposed monolayer of endotoxin. To combat *A. baumannii* infections, polymyxins, including polymyxin E (colistin), are used as agents of last resort against multidrug resistant strains [4]. Polymyxins are cyclic cationic peptides produced by gram-positive soil bacteria discovered in 1947 [5]. Colistin was used in the 1950s and was abandoned in most of the part of the world in the early 1980s because it was regarded as nephrotoxic [6]. The lack of new antibiotic discovery associated with the emergence of multidrug-resistant bacteria led to a regain of interest in colistin molecules [7]. The antibacterial activity of colistin relies on the endotoxin or lipopolysaccharide (LPS) interaction and the disruption of the bacterial outer and inner membranes cells [8,9,10] but with a mechanism of action not perfectly understood [11]. Three domains compose the LPS, the lipid A integrated in the outer membrane, the core oligosaccharide, and the O antigen. However, mucosal pathogens including *A. baumannii* often lack the O antigen and produce instead an extended core oligosaccharide. This endotoxin molecule is termed lipooligosaccharide (LOS) [12].

However, colistin overuse has led many gram-negative bacteria to develop several different resistance strategies such as loss or modifications of the LPS or LOS [13,14,15]. For example, a LOS-deficient colistin-resistant strain of *A. baumannii* showed changes in the membrane potential probably due to the impact on the colistin affinity of the outer membrane modification [16]. In *A. baumannii* LOS, modifications correspond mainly to the addition of galactosamine [17] or phosphoethanolamine (pEtN) moiety on lipid A. The enzyme EptA (also known as PmrC or LptA in *Neisseria meningitidis*) and MCR-1/2 are chromosome- and plasmid-encoded, respectively; they catalyze pEtN addition in many gram-negative bacteria [18,19,20]. Moreover, mutations in genes involved in LPS synthesis are often associated with colistin resistance [21,22]. The main substrate needed to transfer pEtN to LOS is the glycerophospholipid phosphoethanolamine (PE) [23], but how the colistin resistance impacts the lipid composition of the bacterial membrane is not well understood. Nonetheless, in several bacterial species, it is known that lipid composition of the membrane is correlated with antibiotic resistance [24,25,26,27,28,29]. The transcriptomic responses of *A. baumannii* to colistin highlighted significant lipid-related genes reenforcing the changes induced by this antibiotic on bacterial membrane [30]. Only a few studies with modern lipidomic approaches analyzed the alteration of lipid profiles in polymyxin-resistant bacteria [31,32,33]. For *A. baumannii*, one work analyzed key metabolites between polymyxin-susceptible and polymyxin-resistant strains, including some glycerolipids [32]. Shortly after, the most complete lipidomic study using combined MALDI-TOF/MS and TLC analyses was done on the strain ATCC19606^T^ isolated in 1948 [34], long before the emergence of MDR strains and without antibiotic resistance [35]. 

The goal of the present study is to explore the impact of colistin resistance on the lipid composition of *A. baumannii* recently isolated from patients. We performed state-of-the-art liquid chromatography-high-resolution tandem mass spectrometry (LC-HRMS^2^)-based on untargeted lipidomics to improve our understanding of the mechanism of action of colistin resistance. Here, nine lipid sub-classes were identified, (PE), phosphatidylglycerol (PG), lysophosphatidylethanolamine (LPE), hemibismonoacylglycerophosphate (HBMP), cardiolipin (CL), monolysocardiolipin (MLCL), phosphatidylcholine (PC), and the glycerolipids, diacylglycerol (DAG), and triacylglycerol (TAG). For many of these lipids, the fatty acid composition is identified. When the colistin susceptible strain was grown with colistin, the palmitic acid content was upregulated. Inversely, for resistant and susceptible strains, PE significantly decreased. The most abundant lipid species are PE 34:1, PG 34:1, HBMP 50:2, CL 68:2, MLCL 52:2, PC 32:1, and TAG 50:1. These species are among the upregulated lipids when the bacteria are grown with colistin. Our findings describe the plasticity of the bacterial membranes during colistin treatment and show a significant effect for different glycerolipids. Therefore, understanding the mechanism(s) that bacteria develop to defeat antibiotics, and the lipids biosynthesis in particular, should lead to the development of new therapies [36,37]. 

## 2. Results

### 2.1. Colistin Susceptibility

Detection of colistin resistance for *A. baumannii* relies on minimal inhibition concentration (MIC). This method was chosen as the unique reference by the working group, including the Clinical and Laboratory Standards Institute (CLSI) and the European Committee on Antimicrobial Susceptibility Testing (EUCAST), with a concentration of 2 µg/mL of colistin required to kill susceptible bacteria [38]. A colistin-susceptible strain of a clinical isolate of *A. baumannii* denominated 721164 with acquired resistance to β-lactam with β-lactamase (AmpC), carbapenemase (OXA-23), and aminoglycoside resistance methylase (ArmA) was selected. After cultivation by gradually increasing colistin concentration from 0 to 5 μg/mL, a colistin resistant population was isolated. The native (susceptible) and the isolated (resistant) populations were tested by MIC at 0.5 and >128 µg/mL colistin concentrations (Table 1).

### 2.2. Lipids Annotated in Acinetobacter baumannii 721164 by LC-HRMS^2^

Lipid extraction was performed on harvested cells of the susceptible strain grown until the stationary phase was reached (16 h of growth) and analyzed by a liquid chromatography coupled with a hybrid quadrupole time-of-flights (QTOF). In negative mode, 87 glycerophospholipids were annotated, belonging to six lipid sub-classes: PE, PG, LPE, HBMP (also known as Acyl-PG), CL, and MLCL. In positive mode, 43 glycerolipids were annotated belonging to three lipid sub-classes: PC, DAG, and TAG (Figure 1). The percentage of each lipid species was calculated to identify the most representative ones. For CL and MLCL, PE and LPE, and TAG and DAG, the percentage was calculated by gathering the two sub-classes as the MLCL, LPE, and DAG are intermediates of mainly CL, PE, and TAG, respectively. The most abundant lipids for each sub-class are: PE 34:1, PG 34:1, HBMP 50:2, CL 68:2, MLCL 52:2, PC 32:1, and TAG 50:1 (highlighted in red in Table 2). These results are in good agreement with the results obtained on the strain ATCC19606^T^ by MALDI-TOF/MS in the negative ion mode, but our analyses allow to annotate more lipid species. Furthermore, the analysis of the lipid extract by positive ion mode appends three lipid sub-classes in the *A. baumannii* lipidome [35]. For CL, 32 species were identified, 12 MLCL, 12 HBMP, 5 PC, 3 LPE, 15 PE, 13 PG, 11 DAG, and 27 TAG.

### 2.3. Colistin and Lipid Profiles of Acinetobacter baumannii by LC-HRMS^2^

Comparative untargeted lipidomics were employed to identify differences between colistin-susceptible and colistin-resistant *A. baumannii* grown with or without colistin at non-lethal concentrations. The colistin-susceptible strain was cultivated with 0 µg/mL, 0.125 µg/mL, or 0.25 µg/mL of colistin. The colistin-resistant strain was cultivated with 0 µg/mL, 5 µg/mL. A principal component analysis (PCA) scores plot was used to highlight overall variances between the two strains cultivated at different concentrations of colistin using normalized values by employing the internal standard EquiSPLASH (Figure 1). Each sample is shown as a single point with a 95% confidence interval ellipse representing the range of the data. The colistin-resistant strain grown with 5 µg/mL of colistin overlaps with all other conditions. The colistin-susceptible strain grown with 0.25 µg/mL does not overlap with any growth conditions with this strain, indicating that these treatments affect at least in part of the *A. baumannii* lipid content. Among each lipid sub-class, PG has the largest confidence ellipses only for the colistin-susceptible and resistant strains grown without antibiotics, while the two strains cultivated with antibiotics overlapped. With less confidence, similar results were observed for CL/MLCL and TAG/DAG. PCA for PE/LPE and HBMP were the most dramatically affected by 0.25 µg/mL of colistin and between the two conditions of the colistin-resistant strain. The susceptible and the resistant strains without antibiotics (Figure 1, respectively, in purple and red) are overlapping for HBMP and to a less extent for all lipids and PE/LPE. These results indicate that the lipid content of the susceptible and resistant Acinetobacter strains is already affected in the absence of colistin.

The fatty acid content of the colistin-susceptible strain (Figure 2A) and colistin-resistant strain (Figure 2E) grown with different colistin concentrations was evaluated by gas chromatography with a flame ionization detector (GC-FID). Since C12:0 was not always detected, we decided not to represent this fatty acid in the figure. For the colistin-susceptible strain, a significantly higher quantity of palmitic acid (C16:0) was observed proportionally with an antibiotic concentration increase (Figure 2A). Based on the lipidomic data, the percentages of the glycerophospholipids annotated in negative ion mode (PE, PG, and LPE) were calculated and compared. PE significantly decreased concomitantly with an increase of PG for the colistin-susceptible strain (Figure 2B), whereas no significant differences were observed for CL, MLCL, and HBMP (Figure 2C). At the same time, glycerolipids annotated in the positive ion mode (DAG, PC and TAG) were significantly affected. TAG percentage was higher when incubated with 0.25 µg/mL and is associated with a decrease of PC (Figure 2D). Fatty acid analysis was also performed for the colistin-resistant strain grown with and without colistin (Figure 2E). Unlike the colistin-susceptible strain, no significant differences were observed for C16:0. Regarding the different lipid sub-classes, the same effect was observed on PE and PG (Figure 2F). However, a significant decrease of CL and an increase of HBMP were observed when growing the strain with antibiotics (Figure 2G) but with no effect for DAG, PC, or TAG (Figure 2H).

We next examined the individual lipid species involved in the bacterial adaptation in *A. baumannii* under colistin growth (Figure 3). The highest abundant species of PE and PG (PE 34:1 and PG 34:1) are upregulated for both susceptible and resistant strains grown with colistin. Most of the other species are within 5% of the total lipid sub-classes associated with a few abundant species. Eight species are upregulated at the highest concentration of colistin for the susceptible strain and for the resistant strain cultivated with colistin. These shared species are monounsaturated and include PE 32:1, PE 34:1, HBMP 48:1, HBMP 50:1, TAG 48:1, TAG 50:1, PG 34:1, and PG 32:1. Among them, only HBMP 48:1 is below 5%.

## 3. Discussion

Antibiotic resistance and lipid content adaptation of different bacterial species was reported for several bacteria including *Staphylococcus aureus*, *Escherichia coli* (*E. coli*), and *Pseudomonas aeruginosa* (*P. aeruginosa*) [25,27,28,31]. Understanding how bacteria overcome antibiotic treatment and notably the lipids biosynthesis could lead to the development of new therapies [36].

Gram-negative bacteria are enveloped by two lipid bilayers, separated by a periplasmic space containing a peptidoglycan cell wall. The inner and outer membranes are composed of glycerophospholipids. *A. baumannii* is composed of PE, PG, PC, CL, MLCL, HMBP, and LPE lipids. A recent study using 2D thin-layer chromatography on the inner and outer membranes of *A. baumannii* clearly identified PE, PG, and CL with three species of PE and PG corresponding to the 32:1, 34:1, and 36:2 of each sub-class. PE 34:1 and PG 34:1 were identified as the most abundant [39]. These results are in good agreement with our identification of the most abundant PE and PG lipids. The most complete work to date found 38 different species but without data concerning the fatty acids content of each species [35]. In the current study, we annotated and quantified 130 lipid species including TAG and DAG, reserve compounds widespread among bacteria [40]. Already detected in *A. baumannii* but with a composition not fitting the nine fatty acids found in this bacterium (C12:0, C14:0, C15:0, C16:0, C16:1, C17:0: C17:1, C18:0, and C18:1) with structures such as TAG 46:5 and TAG 64:13 [32]. While TAG with 46 fatty acids carbons was annotated in our analysis, five unsaturations were not compatible with the fatty acid content. Likewise, neither 13 unsaturations could be found in this bacterium nor 64 fatty acids carbons. Other glycerophospholipids detected in previous studies [32,35] such as phosphatidic acid (PA), phosphatidylserine (PS), lysophosphatidylglycerol (LPG), hydroxylated-PE, CL, and MLCL (PE-OH, CL-OH, and MLCL-OH) were not detected in our conditions. However, phosphatidylmethanol (PMeOH) lipids were annotated but, since these minor lipids are formed during lipid extraction, they were not considered [41]. Interestingly, PC was annotated with six different species. Despite being a major lipid in eukaryotic cells, this glycerophospholipid is estimated to be present in about 15% of the bacteria and often associated with pathogenic host-microbe interactions. The relative amount of PC can vary from a few percent for *P. aeruginosa* to up to 70% for *Acetobacter aceti* [42]. For *A. baumannii*, based on this study it seems to account for an amount below 1%. Bacteria can employ two pathways to synthesize PC, by using choline as a substrate via PC synthases or through PE methylation [43]. Homologs of PC synthase were found in *A. baumannii* by using the amino acid sequences from *P. aeruginosa* as query, which consolidate this observation (data not shown).

Trent and coworkers performed the analysis of LOS-deficient *A. baumannii* strains in comparison with a wild-type strain using the ^32^P-radiolabeled standard to quantify glycerophospholipids by TLC on PE, PG, and CL [44]. In the same work, a LC-MS/MS analysis for PE, PG, CL, LPE, and PA showed no notable differences in the ratio of glycerophospholipid structures. Furthermore, a previous report described for a colistin-resistant LOS-deficient *A. baumannii* an enrichment of glycerophospholipids with shorter fatty acids [32] associated with a slower growth rate because of complete loss of LOS [15]. Herein, neither shorter fatty acids nor a slower growth rate (data not shown) was found associated with colistin resistance, suggesting a correlation between LOS deficiency and shorter fatty acids in glycerophospholipids but not with colistin resistance. Nevertheless, these same authors described two key metabolites linked with PE (ethanolamine phosphate and glyceroethanolamine phosphate) that were significantly lower in abundance for *A. baumannii* colistin-resistant strains [32]. This observation could explain at least in part the quantitative decrease in PE observed here when bacteria were grown in the presence of colistin and points out the potential use of PE as a substrate to transfer pEtN moiety on lipid A as described [14,45]. LOS have a negative charge targeted by colistin, and the addition of pEtN helps *A. baumannii* to evade the colistin action by reducing the net-negative charge of the outer membrane.

Interestingly, this observation occurs for colistin-susceptible and resistant strains, suggesting a rapid adaptation of the bacteria to a sublethal concentration of colistin. While shorter fatty acids were not observed for colistin-resistant bacteria, a significant increase in palmitic acid was noticed for the colistin-susceptible strain grown with a low concentration of colistin. It was long established that bacteria can modify their fatty acid composition during antimicrobial treatment, which can affect bacterial susceptibility to antimicrobials [46,47,48]. Saturated fatty acids are hypothesized to limit pore-forming antimicrobial activity by lowering bacterial membrane fluidity [49,50]. Likewise, in the presence of colistin, susceptible *A. baumannii* could lower membrane fluidity to circumvent disruption of the outer membrane induced by colistin at a sublethal concentration. Since PE is known to be a regulator of membrane fluidity in most eukaryotic cells [51] and bacteria [52], the palmitic acid increase could compensate the impact of the PE decrease on the membrane fluidity. A recent study on *E. coli* highlighted the influence of PE on the distribution of other lipid sub-classes between leaflets by vectorial molecular probes [52]. By using the same approach, PE asymmetry and homeostasis under colistin treatment could be better characterized.

Gram-negative bacteria have an asymmetric lipid distribution, essential for their viability [39]. The addition of a sublethal concentration of colistin for *A. baumannii* 721164 resulted in an adaptation mechanism for the cells to maintain membrane structure and function. This work will help the lipid analysis of *A. baumannii* and, in more general terms, the understanding of the colistin effect on gram-negative bacteria. Obviously, further studies are necessary on more *A. baumannii* strains to elucidate the physiological consequences of our findings. Therefore, by using state-of-the-art LC-HRMS^2^ based on untargeted lipidomics, these results should also help future works focused on the analyses of outer and inner membranes. Deciphering the plasma membrane homeostasis of *A. baumannii* under environmental stresses will help to design new antimicrobial strategies.

## 4. Conclusions

Altogether, our present observations show that exposure of a clinical isolate of *A. baumannii* to sublethal and lethal amounts of colistin triggers a set of lipid changes. Palmitic acid percentage increases only for the susceptible strain, suggesting an adaptation of the plasma membrane fluidity to limit colistin membrane destabilization. PE percentage decreases significantly on both strains, which, in turn, provides pEtN for LOS modification. As described in the literature [53], this addition of pEtN moiety on lipid A reduces the overall net-negative charge of the bacterial surface and confers resistance to colistin. The present study gives us an interesting view on lipid homeostasis of one clinical isolate. The 130 lipid species described could help future works dealing with *A. baumanni* lipidomics in general and, more specifically, on inner and outer membrane asymmetry. Our efforts could contribute to the development of novel inhibitory molecules targeting bacterial lipids to fight the difficult-to-treat diseases associated with *A. baumannii*.

## 5. Materials and Methods

### 5.1. Bacterial Strains and Growth Conditions

*A. baumannii* 721164 (AmpC, OXA-23, ArmA), a clinical isolate obtained from the microbiology department of the hospital of Valenciennes, was grown with aeration at 180 rpm in lysogeny broth (LB) Miller at 37 °C. No heteroresistance for colistin was detected for this strain by using population analysis profiling, as described previously [54]. The colistin-resistant *A. baumannii* 721164 was selected by gradually increasing colistin concentration from 0 to 5 μg/mL in 5 mL LB Miller. The first inoculation was 5 × 10^5^ CFU/mL without colistin. 1 mL of this suspension was transferred into a new tube containing 4 mL LB Miller with a 0.125 µg/mL for 24 h. The obtained bacteria were transferred into a new tube of 4 mL LB Miller but with a final concentration of 0.25 µg/mL for the same incubation time. The same treatments were applied sequentially with 0.5, 1, 2, and 5 µg/mL of colistin to the treated cells. In total, six passages were performed.

### 5.2. Susceptibility Testing to Colistin

The minimal inhibitor concentration (MIC) of colistin against *A. baumannii* was measured with broth microdilution (BMD) according to the Clinical Laboratory Standard Institute (CLSI) and European Committee on Antimicrobial Susceptibility Testing (EUCAST) guidelines with some modifications (http://www.eucast.org/clinical_breakpoints/, accessed on 30 April 2021) [55]. BMD panels were prepared in sterile flat polystyrene 96-well microplates (P7491-1CS from Sigma Aldrich, Saint-Quentin Fallavier, France). A 1280 μg/mL stock solution of colistin sulfate (PHR1605-1G from Sigma Aldrich, Saint-Quentin Fallavier, France) was prepared in sterile Milli-Q (MQ) water. Incremental dilutions were made in a LB Miller medium to a final concentration of 5 × 10^5^ CFU/mL. Two-fold dilutions of colistin concentrations were tested and ranged from 0.125 to 128 μg/mL. For all the strains, eight replicates of 100 μL/well were incubated for 24 h at 37 °C. The OD was measured at 600 nm on a plate reader. The interpretations are based on CLSI that provided susceptibility breakpoints for colistin against *A. baumannii*: MIC ≤ 2 μg/mL (susceptible), MIC ≥ 4 μg/mL (resistant).

### 5.3. Lipid Extraction

The lipids were extracted by using the Bligh and Dyer method with methanol (MeOH)/chloroform (CHCl_3_)/H_2_O 2:1:0.8 *v*/*v*/*v* [56]. Briefly, *A. baumannii* was inoculated at an initial concentration of 5 × 10^5^ CFU/mL and grown in 100 mL of LB Miller medium at different concentrations of colistin (0, 0.125, and 0.25 μg/mL (susceptible) or 0, 0.5 µg/mL (resistant)) at 37 °C overnight on an orbital shaker with 180 rpm. Each condition was repeated three times. Cells were harvested by centrifugation at 3500× *g* for 5 min at 4 °C, washed twice in MQ water. The cell pellet was resuspended in 1ml MQ water in a 50 mL centrifuge tube and left on ice for 5–10 min. Then, 4 mL MeOH and 2 mL CHCl_3_ were added and vortexed for 5 min. After incubating on ice for 10 min, 4 mL of CHCl_3_ and 2 mL of MQ water were added and shaken for 3 min. The lower phase was collected and supplemented with 1 mL of 0.5 M sodium chloride (NaCl). After hand-shaking for 1 min, phase separation was enhanced by centrifugation at 1500× *g* for 5 min. The final lower phase, approximately 4 mL, was collected and dried with nitrogen (N_2_). The lipids were stored at −20 °C until needed for further analyses.

### 5.4. Fatty Acids Analysis by GC-FID

The extracted lipids were suspended in 1 mL of CHCl_3_. Then, 200 μL of this solution was transferred to a new glass tube and dried under a stream of nitrogen (N_2_) at room temperature. Then, 1 mL of freshly prepared 5% H_2_SO_4_ (*v*/*v*) in MeOH, containing 50 μL of butylated hydroxytoluene (BHT; 1 mM), were added and completed with 300 μL of toluene. The mix was vortexed vigorously for 30 s before being heated for 1 h at 85–90 °C. Fatty acid methyl esters (FAME) were extracted with 1.5 mL of 0.9% NaCl (*w*/*v*) and 1 mL of heptane. The tube was vortexed and then centrifuged briefly to facilitate phase separation. The heptane extracts (upper organic phase) were transferred to a new glass tube and evaporated under a stream of N_2_, then dissolved in 50 μL heptane. FAME composition was determined by GC-FID. The GC-FID analyses were carried out on a Shimadzu GC 2010 plus gas chromatography, and chromatographic separation was performed on a BPX70 column (60 m × 0.25 mm id, film thickness 0.25 μm). The GC parameters were the following: the oven temperature program was 120 °C to 250 °C at 10 °C/min with helium as carrier gas. One µL of sample was injected in a 250 °C inlet with a 40:1 split ratio. Methyl ester derivatives were detected using a FID at 280 °C.

### 5.5. LC-HRMS^2^ Analyses

The extracted lipids were resuspended in 200 µL of isopropanol. LC was performed based on a modified protocol, as described before [57]. Briefly, liquid chromatograph used a Waters Aquity UPLC C18 column (100 × 2.4 mm, 1.7 µm) coupled to an Acquity UPLC CSH C18 VanGuard precolumn (5 × 2.1 mm; 1.7 µm) at 65 °C. Mobile phases were 60:40 (vol/vol) acetonitrile/water (solvent A) and 90:10 (vol/vol) isopropanol/acetonitrile (solvent B). For the positive mode, mobile phases were buffered with 10 mM ammonium formate and 0.1% formic acid. For the negative mode, the mobile phases were buffered with 10 mM ammonium acetate and 0.1% acetic acid. The flow rate was set at 0.6 mL/min and with an injection volume of 2 µL. LC-electrospray ionization (ESI)-HRMS^2^ analyses were achieved by coupling the LC system to a hybrid quadrupole time-of-flight (QTOF) mass spectrometer Agilent 6538 (Agilent Technologies) equipped with dual electrospray ionization (ESI), as described before in [58]. The source temperature, fragmentor, and the skimmer were set up at 350 °C, 150 V, and 65 V, respectively. The acquisition was made in full scan mode between 100 *m/z* and 1700 *m*/*z*, with a scan of 2 spectra per second. Selected parent ions were fragmented to a collision energy of 35 eV. MS2 scans were performed on the sixth-most intense ions. Two internal reference masses were used for in-run calibration of the mass spectrometer (121.0509 and 922.0098 in positive-ion mode and 112.9856 and 1033.9881 in negative-ion mode). MassHunter B.07 software enabled the control of the parameters of the machine, and acquired and processed the data. The mass spectra were acquired in positive- and negative-ion modes. Internal standards were used to quantify lipid classes in positive and negative ion modes. Then, 2 µL of internal standards (EquiSPLASH LIPIDOMIX, 330731-1EA from Sigma Aldrich, Saint-Quentin Fallavier, France) were added prior to extraction. Among the 13 deuterated lipids present in the internal standard, CL, HBMP, and MLCL were not present and, deuterated PGs were used as these lipids shared the same head group.

### 5.6. Data Processing and Annotation

Agilent generated files (*.d) were converted to the *.mzML format using MSConvert [59]. File (*.mzML) data sets were processed using MZmine [60]. Data were processed with the software MS-DIAL version 4.38 [61]. Baseline correction, peak detection, alignment, gap filling, and adduct identification for raw data were performed with MS-DIAL, according the parameters described in [61]. Lipids were annotated according to *m*/*z*, retention time, and MS/MS spectra obtained from MS-DIAL. Peak height was used as the mass spectral intensity for each annotated lipid. The nomenclature for lipid sub-class follows the definition from [62].

### 5.7. Statistical Analysis

MetaboAnalyst 5.0 [63] was used to estimate variation across the sample group (PCA and Heat map). For each sample, the peak area of each lipids was normalized to the total peak areas of all lipids. Significance was analyzed using ANOVA, and Tukey’s HSD was used as a post hoc test. Graphs were made using Prism Software V 5.0 (https://www.graphpad.com/support/prism-5-updates/, accessed on 31 January 2021); statistical significance was evaluated with student’s *t* test or a one-way analysis of variance. The results were considered significant for a *p* value of ≤0.05. Three assays were carried out for lipid analysis.

## Figures and Tables

**Figure 1 antibiotics-10-00528-f001:**
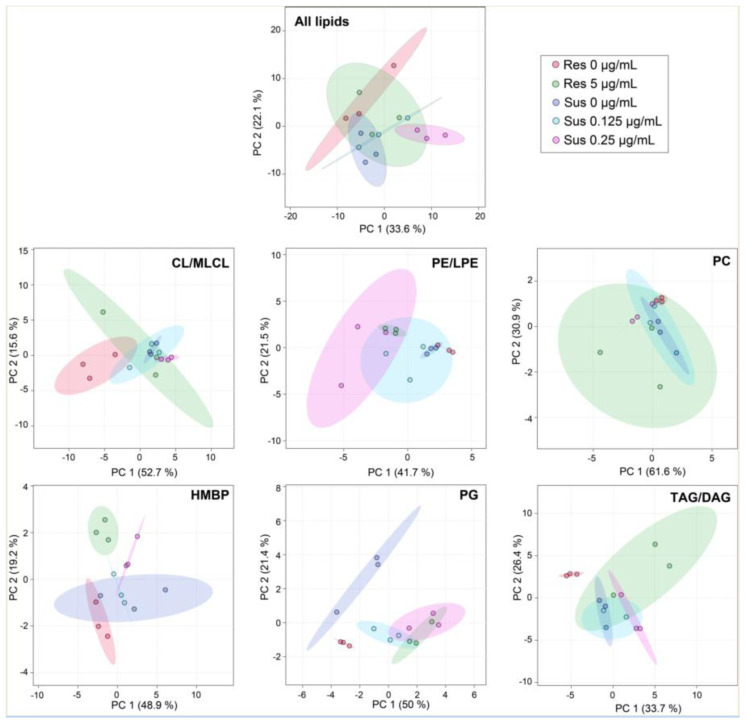
Low concentrations of colistin alter *Acinetobacter baumannii* lipid profiles. PCA scores plot showing variances in lipid species between susceptible strains without and with colistin at 0.125 µg/mL or 0.25 µg/mL and resistant without or with colistin at 5 µg/mL of *Acinetobacter baumannii* 721164. The results correspond to n = 3 biologically independent samples. Ellipses represent 95% confidence intervals. The analysis was performed using MetaboAnalyst V5.0 (https://www.metaboanalyst.ca/, accessed on 15 January 2021).

**Figure 2 antibiotics-10-00528-f002:**
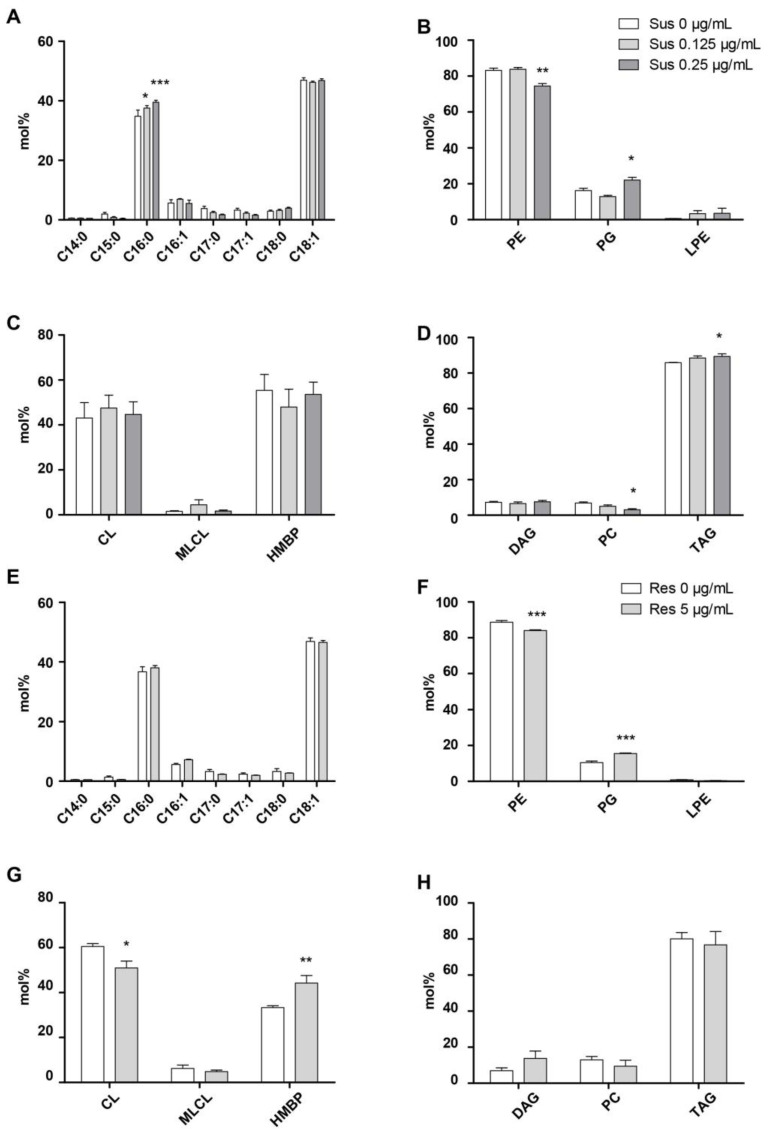
*A. baumannii* 721164 lipid content differs when incubated with colistin. (**A**) Fatty acid methyl ester analysis of *A. baumannii* 721164 by GC-FID cultivated without or with colistin at 0.125 µg/mL or 0.25 µg/mL. (**B**–**D**) mol % of different lipid species analyzed by LC-MS. (**E**) Fatty acid methyl ester analysis of *A. baumannii* 721164 resistant to colistin by GC-FID cultivated without or with colistin at 5 µg/mL. (**F**–**H**) mol % of different lipid species analyzed by LC-MS. The results correspond to n = 3 biologically independent samples. Statistical significances were determined by a two-tailed student’s *t* test ***, *p* ≤ 0.001; **, *p* ≤ 0.005; *, *p* ≤ 0.05.

**Figure 3 antibiotics-10-00528-f003:**
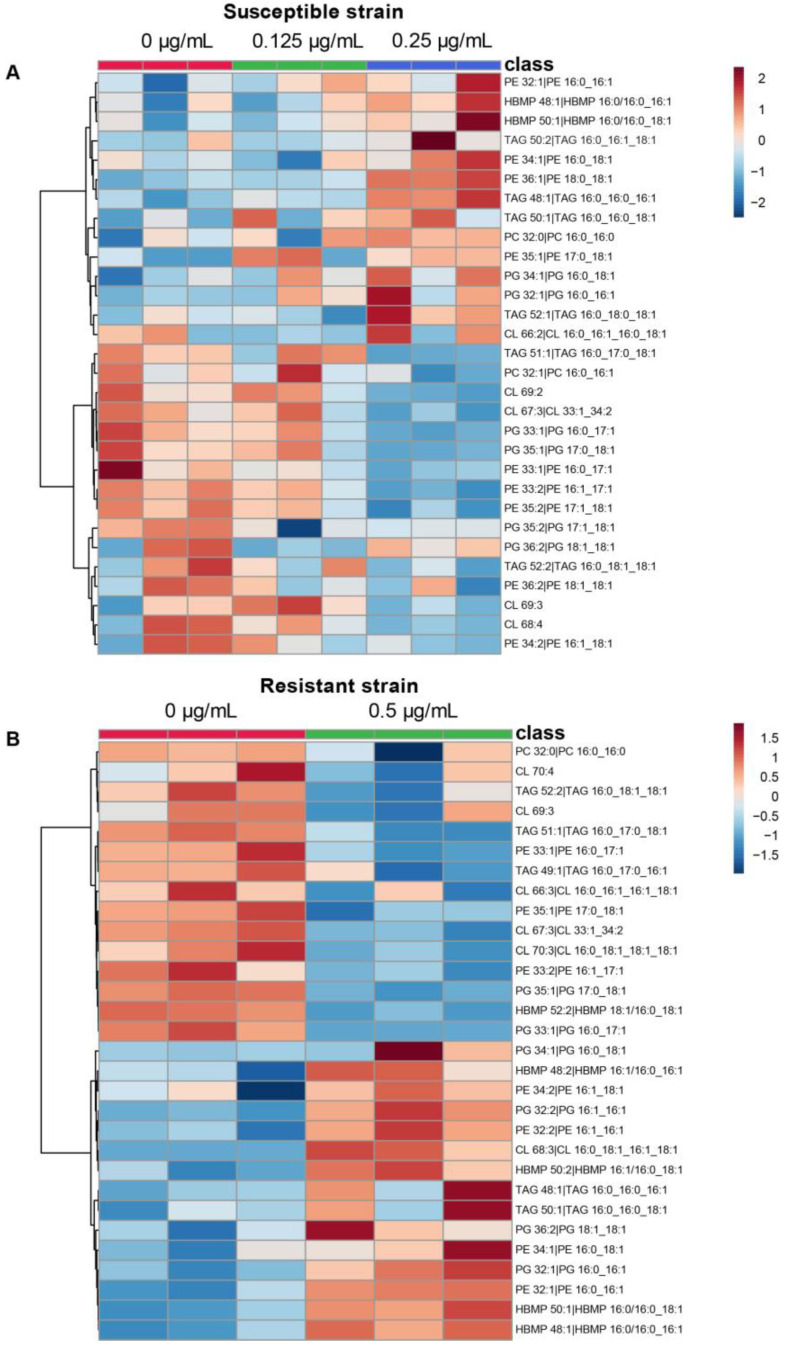
Colistin treated strains and lipid species. Heat map of the 30 most statistically different lipid species analyzed by LC-MS. (**A**) Comparison of *A. baumannii* 721164 susceptible strain at 0 (red), 0.125 µg/mL (green), and 0.25 µg/mL (blue) colistin. (**B**) Comparison of *A. baumannii* 721164 resistant without (red) or with 0.5 µg/mL (green) colistin, respectively. The results correspond to n = 3 biologically independent samples. Color coding indicates greater deviation from the mean from all samples for a particular lipid. The analysis was performed using MetaboAnalyst V5.0.

**Table 1 antibiotics-10-00528-t001:** Minimal inhibitory concentration (MIC) of colistin of the investigated *A. baumannii* 721164 determined by broth microdilution.

*Acinetobacter baumannii* 721164	Susceptible	Resistant
Colistin	0.5	>128

Note: MIC values are in µg/mL.

**Table 2 antibiotics-10-00528-t002:** Lipid assignments of the total lipid extract of *Acinetobacter baumannii* 721164 grown in LB and analyzed by LC-HRMS^2^. The adduct type for Cardiolipin (CL), Monolysocardiolipin (MLCL), Hemibismonoacylglycerophosphate (HBMP), Lysophosphatidylethanolamine (LPE), Phosphoethanolamine (PE), and Phosphatidylglycerol (PG) is [M-H]-; for Phosphatidylcholine (PC) is [M+H]+; for Diacylglycerol (DAG) and Triacylglycerol (TAG) is [M+NH4]+. Lipids in grey are found in quantities over 5% of the total lipid sub-classes with CL/MLCL, PE/LPE, and DAG/TAG gathered to calculate the percentage of each lipid species. The highest abundant lipid species are in red.

Lipids	*m*/*z*	Retention Time	Assignment
Cardiolipin (CL)	1293.89026	9.502	CL 60:1
	1291.87561	9.099	CL 60:2
	1289.85425	8.62	CL 60:3
	1305.89014	9.326	CL 61:2
	1303.8728	8.866	CL 61:3
	1321.9187	9.991	CL 62:1
	1319.90637	9.495	CL 62:2
	1317.89062	9.138	CL 62:3
	1335.93335	10.148	CL 63:1
	1333.92212	9.805	CL 63:2
	1331.90564	9.362	CL 63:3
	1347.93762	10.034	CL 64:2|CL 16:0_16:1_16:0_16:1
	1345.9209	9.483	CL 64:3|CL 16:0_16:1_16:1_16:1
	1361.9541	10.21	CL 65:2
	1359.93982	9.837	CL 65:3
	1357.92249	9.389	CL 65:4
	1375.96924	10.205	CL 66:2|CL 16:0_16:1_16:0_18:1
	1373.953	10.067	CL 66:3|CL 16:0_16:1_16:1_18:1
	1371.93896	9.656	CL 66:4
	1389.98438	10.276	CL 67:2
	1387.96814	10.219	CL 67:3|CL 33:1_34:2
	1385.95239	9.863	CL 67:4
	1403.99841	10.299	CL 68:2|CL 16:0_18:1_16:0_18:1
	1401.98572	10.219	CL 68:3|CL 16:0_18:1_16:1_18:1
	1399.96814	10.092	CL 68:4
	1418.01538	10.313	CL 69:2
	1415.99963	10.279	CL 69:3
	1413.98303	10.223	CL 69:4
	1430.01562	10.303	CL 70:3|CL 16:0_18:1_18:1_18:1
	1427.99902	10.266	CL 70:4
	1444.02808	10.317	CL 71:3
	1456.02783	10.303	CL 72:4
Monolysocardiolipin (MLCL)	1107.68982	6.044	MLCL 48:0
	1121.70483	6.354	MLCL 49:0
	1117.67041	8.435	MLCL 49:2
	1135.72205	6.712	MLCL 50:0
	1137.74084	7.17	MLCL 50:2|MLCL 16:0_34:2
	1149.73999	7.012	MLCL 51:0
	1163.75537	7.218	MLCL 52:0
	1165.77185	7.785	MLCL 52:2|MLCL 18:0_34:2
	1157.70374	6.694	MLCL 52:3
	1191.78455	7.806	MLCL 53:0
	1187.75085	7.762	MLCL 53:2
	1185.7345	7.264	MLCL 53:3
Hemibismonoacylglycerophosphate (HBMP)	901.65656	7.196	HBMP 44:1|HBMP 16:1/12:0_16:0
	929.68842	7.81	HBMP 46:1|HBMP 16:0/14:0_16:1
	927.67273	7.244	HBMP 46:2|HBMP 14:0/14:1_18:1
	957.72144	8.389	HBMP 48:1|HBMP 16:0/16:0_16:1
	955.70532	7.872	HBMP 48:2|HBMP 16:1/16:0_16:1
	953.69006	7.32	HBMP 48:3|HBMP 16:1/16:1_16:1
	970.72339	8.17	HBMP 49:3
	985.75183	8.934	HBMP 50:1|HBMP 16:0/16:0_18:1
	983.73761	8.36	HBMP 50:2|HBMP 16:1/16:0_18:1
	981.72137	7.924	HBMP 50:3|HBMP 16:1/16:1_18:1
	1000.76843	9.183	HBMP 51:2
	1011.7688	8.998	HBMP 52:2|HBMP 18:1/16:0_18:1
Phosphatidylcholine (PC)	734.5694	6.325	PC 32:0|PC 16:0_16:0
	732.55377	6.661	PC 32:1|PC 16:0_16:1
	730.53809	4.832	PC 32:2|PC 16:1_16:1
	760.58508	6.182	PC 34:1|PC 16:0_18:1
	758.5694	5.525	PC 34:2|PC 16:1_18:1
Lysophosphatidylethaniolamine (LPE)	450.26407	1.067	LPE 16:1
	464.28006	1.321	LPE 17:1
	478.29681	1.648	LPE 18:1
Phosphatidylethanolamine (PE)	662.479	5.642	PE 30:0|PE 14:0_16:0
	660.46484	4.993	PE 30:1|PE 14:0_16:1
	674.48035	5.368	PE 31:1|PE 15:0_16:1
	690.5108	6.406	PE 32:0|PE 16:0_16:0
	688.49481	5.736	PE 32:1|PE 16:0_16:1
	686.47961	5.093	PE 32:2|PE 16:1_16:1
	702.51129	6.114	PE 33:1|PE 16:0_17:1
	700.49573	5.446	PE 33:2|PE 16:1_17:1
	716.52686	6.519	PE 34:1|PE 16:0_18:1
	714.51105	5.83	PE 34:2|PE 16:1_18:1
	730.54224	8.1	PE 35:1|PE 17:0_18:1
	730.54279	6.858	PE 35:1|PE 17:0_18:1
	728.52765	6.2	PE 35:2|PE 17:1_18:1
	744.55786	7.261	PE 36:1|PE 18:0_18:1
	742.54224	6.598	PE 36:2|PE 18:1_18:1
Phosphatidylglycerol (PG)	693.47375	4.525	PG 30:0|PG 14:0_16:0
	691.45935	4.051	PG 30:1|PG 14:0_16:1
	705.47412	4.327	PG 31:1|PG 15:0_16:1
	719.49042	4.616	PG 32:1|PG 16:0_16:1
	717.47479	4.155	PG 32:2|PG 16:1_16:1
	733.50592	4.935	PG 33:1|PG 16:0_17:1
	731.49005	4.407	PG 33:2|PG 16:1_17:1
	747.52142	5.227	PG 34:1|PG 16:0_18:1
	745.50616	4.751	PG 34:2|PG 16:1_18:1
	761.53699	5.61	PG 35:1|PG 17:0_18:1
	759.52203	5.016	PG 35:2|PG 17:1_18:1
	775.55298	5.948	PG 36:1|PG 18:0_18:1
	773.53888	5.322	PG 36:2|PG 18:1_18:1
Diacylglycerol (DAG)	586.54303	7.577	DAG 32:0|DAG 16:0_16:0
	584.53033	6.908	DAG 32:1|DAG 16:0_16:1
	582.51422	6.23	DAG 32:2|DAG 16:1_16:1
	598.54559	7.278	DAG 33:1|DAG 16:0_17:1
	614.57684	8.293	DAG 34:0|DAG 16:0_18:0
	612.55981	7.641	DAG 34:1|DAG 16:0_18:1
	610.54663	7.159	DAG 34:2|DAG 16:1_18:1
	626.57391	7.997	DAG 35:1|DAG 17:0_18:1
	642.60565	8.969	DAG 36:0|DAG 18:0_18:0
	640.59149	8.353	DAG 36:1|DAG 18:0_18:1
	638.57568	7.692	DAG 36:2|DAG 18:1_18:1
Triacylglycerol (TAG)	768.71069	10.794	TAG 44:0|TAG 14:0_14:0_16:0
	766.69482	10.333	TAG 44:1|TAG 12:0_16:0_16:1
	796.74341	11.257	TAG 46:0|TAG 14:0_16:0_16:0
	794.72626	10.814	TAG 46:1|TAG 14:0_16:0_16:1
	792.7085	10.435	TAG 46:2|TAG 14:0_16:1_16:1
	810.75525	11.472	TAG 47:0|TAG 15:0_16:0_16:0
	808.74298	11.046	TAG 47:1|TAG 15:0_16:0_16:1
	824.77271	11.622	TAG 48:0|TAG 16:0_16:0_16:0
	822.75842	11.276	TAG 48:1|TAG 16:0_16:0_16:1
	820.74347	10.845	TAG 48:2|TAG 16:0_16:1_16:1
	818.72485	10.461	TAG 48:3|TAG 14:1_16:1_18:1
	838.7934	11.838	TAG 49:0|TAG 16:0_16:0_17:0
	836.77765	11.492	TAG 49:1|TAG 16:0_17:0_16:1
	834.75635	11.105	TAG 49:2|TAG 16:0_16:1_17:1
	852.79919	11.729	TAG 50:0|TAG 16:0_16:0_18:0
	850.78961	11.7	TAG 50:1|TAG 16:0_16:0_18:1
	848.77393	11.296	TAG 50:2|TAG 16:0_16:1_18:1
	846.75836	10.867	TAG 50:3|TAG 16:1_16:1_18:1
	866.8172	11.865	TAG 51:0|TAG 16:0_17:0_18:0
	864.80768	11.853	TAG 51:1|TAG 16:0_17:0_18:1
	862.7915	11.503	TAG 51:2|TAG 16:0_17:1_18:1
	878.82141	11.957	TAG 52:1|TAG 16:0_18:0_18:1
	876.80676	11.706	TAG 52:2|TAG 16:0_18:1_18:1
	874.78894	11.313	TAG 52:3|TAG 16:1_18:1_18:1
	892.83417	12.027	TAG 54:1|TAG 18:0_18:0_18:1
	890.8208	11.855	TAG 54:2|TAG 18:0_18:1_18:1
	906.85333	12.085	TAG 54:3|TAG 18:1_18:1_18:1

## Data Availability

The data presented in this study are available on request from the corresponding author.

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
