# Peer review of "Colistin Treatment Affects Lipid Composition of Acinetobacter baumannii"

_antibiotics, 2021, doi:10.3390/antibiotics10050528_

Round 1

Reviewer 1 Report

This is an important contribution to our understanding of colistin resistance in Acinetobacter, providing an exhaustive list of membrane lipids that may be investigated for their role in colistin resistance. As many mechanisms of colistin resistance involve alterations of the cell surface, this is a valuable dataset. However, it could be analyzed more effectively, particularly in comparisons between the resistant and susceptible strain.

Major Points

There is great insight here in how colistin treatment can alter Acinetobacter lipids in the short term. However, there is no analysis done to address the differences between susceptible and resistant strains.  Comparisons should be done between the susceptible strain and resistant strain without colistin, to show how these adaptations affected lipid content. This would also be complemented by sequencing of the susceptible and resistant strains to determine the genetic changes that occurred to generate resistance, which can then be related to the alterations in lipid content.

More details are needed in how the resistance was evolved in the A. baumannii strain.  How many passages were required?  How long was it grown for?  Additionally, the authors should address if the resistant population was pre-existing (ie. heteroresistance) or if the initial strain was completely susceptible without a resistant subpopulation.

This analysis is done in only a single strain which was evolved to resistance.  Thus many of these results may not be generalizable to other instances of colistin resistance in other A. baumannii strains. While it would be significant work to repeat this for another strain or strains, the authors should address this shortfall in the discussion. This is also another reason why it is imperative to understand the mechanism of resistance in this particular evolved strain.

Minor Points

Lines 141-142: Should probably only reference fig 2A and 2E for the GC-FID.  Also should spell out abbreviation for GC-FID the first time it is mentioned

Lines 147-153 Should specify the groups being discussed here

Line 156 misspelled ‘decrease’

Author Response

This is an important contribution to our understanding of colistin resistance in Acinetobacter, providing an exhaustive list of membrane lipids that may be investigated for their role in colistin resistance. As many mechanisms of colistin resistance involve alterations of the cell surface, this is a valuable dataset. However, it could be analyzed more effectively, particularly in comparisons between the resistant and susceptible strain.

We thank the referee for the positive comments.

Major Points

There is great insight here in how colistin treatment can alter Acinetobacter lipids in the short term. However, there is no analysis done to address the differences between susceptible and resistant strains.  Comparisons should be done between the susceptible strain and resistant strain without colistin, to show how these adaptations affected lipid content. This would also be complemented by sequencing of the susceptible and resistant strains to determine the genetic changes that occurred to generate resistance, which can then be related to the alterations in lipid content.

Answer: We thank the reviewer for these suggestions. More details about the lipid content of the susceptible and resistant strains without colistin have been added from the line 156. “The susceptible and the resistant strains without antibiotics (Figure 1, respectively in purple and red) are overlapping for HBMP and to a less extent for all lipids and PE/LPE. These results indicate that the lipid content of the susceptible and resistant Acinetobacter strains is already affected in the absence of colistin”.

While this is an interesting point, we are not planning to sequence these strains as we do not have access to sequencing resources. However, we have added explanation about genetic adaptation in line 62: “Moreover, mutations in genes involved in LPS synthesis are often associated with colistin resistance [21,22]” and in line 67: “The transcriptomic responses of A. baumannii to colistin have highlighted significant lipid related genes reenforcing the changes induced by this antibiotic on bacterial membrane damage [30]”

More details are needed in how the resistance was evolved in the A. baumannii strain.  How many passages were required?  How long was it grown for?  Additionally, the authors should address if the resistant population was pre-existing (ie. heteroresistance) or if the initial strain was completely susceptible without a resistant subpopulation.

We provided additional details about the heteroresistance and the protocol used for the colistin resistant bacteria in the subsection “Bacterial strains and growth conditions”.

This analysis is done in only a single strain which was evolved to resistance. Thus many of these results may not be generalizable to other instances of colistin resistance in other A. baumannii strains. While it would be significant work to repeat this for another strain or strains, the authors should address this shortfall in the discussion. This is also another reason why it is imperative to understand the mechanism of resistance in this particular evolved strain.

We thank the reviewer for these suggestions. We have added more details starting from line 278. “This work will help lipid analysis of A. baumannii and, in more general terms, the understanding of colistin effect on Gram-negative bacteria. Obviously, further studies are necessary on more A. baumannii strains to elucidate the physiological consequences of our findings”.

Minor Points

Lines 141-142: Should probably only reference fig 2A and 2E for the GC-FID.  Also should spell out abbreviation for GC-FID the first time it is mentioned

The changes were done accordingly.

Lines 147-153 Should specify the groups being discussed here

This information is now included.

Line 156 misspelled ‘decrease’

Change done.

Reviewer 2 Report

Colistin treatment affects lipid composition of Acinetobacter baumannii

The main goal of this work was to analyze the impact of colistin resistance on glycerophospholipds content by using untargeted lipidomics on clinical isolate. Nine lipid sub classes were annotated, including phosphati-dylcholine, rarely detected in bacterial membrane among 130 different lipid species. The manuscript needs a revision to be reconsidered for publication. Therefore, in my point of view, the manuscript could be considered for acceptance but not in its current form. Having said that following revisions are suggested;

Comments:

  1. The abstract is descriptive and qualitative.
  2. Normally an abstract should state briefly the purpose of the study undertaken and meaningful conclusions based on the obtained results. Hence, this needs rewriting. I would expect brief, yet concise, the quantitative data description of the results in the abstract.
  3. The given list of keywords is superficial with broader terms. More specific terms should be used. Replace accordingly.
  4. The introduction is short. More literature should be added with recent and relevant literature.
  5. The novelty of the study should be clearly highlighted in the manuscript at the end of the introduction section, as there are some existing literature reports.
  6. The manuscript should be carefully revised so that the results are better discussed. In my opinion, authors mainly focused on results and the Discussion section lacks scientific depth.
  7. All sections should be critically discussed and compared with the previous reports. This will actually strengthen the manuscript and will highlight the significance of the work.
  8. Conclusion is missing. Herein, I would like to see the major findings and how they are addressing the left behind research gaps and covering current challenges.
  9. Literature needs to be updated with care.

Author Response

The main goal of this work was to analyze the impact of colistin resistance on glycerophospholipds content by using untargeted lipidomics on clinical isolate. Nine lipid sub classes were annotated, including phosphati-dylcholine, rarely detected in bacterial membrane among 130 different lipid species. The manuscript needs a revision to be reconsidered for publication. Therefore, in my point of view, the manuscript could be considered for acceptance but not in its current form. Having said that following revisions are suggested;

We thank the reviewer for the careful review.

Comments:

  1. The abstract is descriptive and qualitative.
  2. Normally an abstract should state briefly the purpose of the study undertaken and meaningful conclusions based on the obtained results. Hence, this needs rewriting. I would expect brief, yet concise, the quantitative data description of the results in the abstract.

The abstract has been revised as requested.

“Multidrug-resistant Acinetobacter baumannii (A. baumannii) causes severe and often fatal healthcare-associated infections due partly to antibiotic resistance. There are no studies on A. baumannii lipidomics of susceptible and resistant strains grown at lethal and sublethal concentrations. Therefore, we analyzed the impact of colistin resistance on glycerolipids content by using untargeted lipidomics on clinical isolate. Nine lipid sub classes were annotated, including phosphatidylcholine, rarely detected in bacterial membrane among 130 different lipid species. The other lipid sub classes detected are phosphatidylethanolamine (PE), phosphatidylglycerol (PG), lysophosphatidylethanolamine, hemibismonoacylglycerophosphate, cardiolipin, monolysocardiolipin, diacylglycerol and triacylglycerol. Under lethal and sublethal concentrations of colistin significant reduction of PE was observed on resistant and susceptible strain, respectively. Palmitic acid percentage was higher at colistin low concentration but only for the susceptible strain. When looking at individual lipid species, the most abundant PE and PG species (PE 34:1 and PG 34:1) are significantly upregulated when the susceptible and the resistant strains are cultivated with colistin. This is to date the most exhaustive lipidomics data compilation of A. baumannii cultivated in presence of colistin. This work is highlighting the plasma membrane plasticity used by this Gram‐negative bacterium to survive colistin treatment”.

3. The given list of keywords is superficial with broader terms. More specific terms should be used. Replace accordingly.

More specific terms were added as follow: Colistin resistance 1; Untargeted lipidomics 2; Acinetobacter baumannii isolate3; Glycerolipids and fatty acids 4

4. The introduction is short. More literature should be added with recent and relevant literature.

More references were added. From line 62: “Moreover, mutations in genes involved in LPS synthesis are often associated with colistin resistance [21,22]”. In line 67 more up to date papers were added.

Other references were cited in the introduction in order to answer to the remarks made by the reviewer 1.

5. The novelty of the study should be clearly highlighted in the manuscript at the end of the introduction section, as there are some existing literature reports.

The novelty of this work has been highlighted in the introduction as follow (line 77): “The goal of the present study is to explore the impact of colistin resistance on the lipid composition of A. baumannii recently isolated from patients. We performed state-of-the-art liquid chromatography-high-resolution tandem mass spectrometry (LC-HRMS2)-based untargeted lipidomics to improve our understanding of the mechanism of action of colistin resistance. Here nine lipid sub classes were identified (PE), phosphatidylglycerol (PG), lysophosphatidylethanolamine (LPE), hemibismonoacylglycerophosphate (HBMP), cardiolipin (CL), monolysocardiolipin (MLCL), phosphatidylcholine (PC) and the glycerolipids, diacylglycerol (DAG) and triacylglycerol (TAG). For many of these lipids the fatty acids composition is identified. When the colistin susceptible strain was grown with colistin the palmitic acid content was upregulated. Inversely, for resistant and susceptible strains, PE significantly decreased. The most abundant lipid sub classes are PE 34:1, PG 34:1, HBMP 50:2, CL 68:2, MLCL 52:2, PC 32:1 and TAG 50:1. These species are among the upregulated lipids when the bacteria are grown with colistin. Our findings describe the plasticity of the bacterial membranes during colistin treatment and show a significant effect for different glycerolipids. Therefore, understanding the mechanism(s) that bacteria develop to defeat antibiotics and, the lipids biosynthesis in particular, should lead to the development of new therapies [36,37]”.

6. The manuscript should be carefully revised so that the results are better discussed. In my opinion, authors mainly focused on results and the Discussion section lacks scientific depth.

The results are now more thoroughly discussed as follow:

From line 215: “A recent study using 2D thin-layer chromatography on the inner and outer membranes of A. baumannii has clearly identified PE, PG and CL with three species of PE and PG corresponding to the 32:1, 34:1 and 36:2 of each sub class. PE 34:1 and PG 34:1 were identified as the most abundant [39]. These results are in good agreement with our identification of the most abundant PE and PG lipids”.

From line 254: “This observation could explain at least in part the quantitative decrease of PE observed here when bacteria were grown in presence of colistin and points out the potential use of PE as a substrate to transfer pEtN moiety on lipid A as described [14,45]. LOS have a negative charge targeted by colistin and the addition of pEtN helps A. baumannii to evade the colistin action by reducing the net-negative charge of the outer membrane”.

From line 267: “Saturated fatty acids are hypothesized to limit pore-forming antimicrobial activity by lowering bacterial membrane fluidity [49,50]. Likewise, in the presence of colistin, susceptible A. baumannii could lower membrane fluidity to circumvent disruption of the outer membrane induced by colistin at sub lethal concentration. Since PE is known to be a regulator of membrane fluidity in most eukaryotic cells [51] and bacteria [52], the palmitic acid increase could compensate the impact of the PE decrease on the membrane fluidity. A recent study in E. coli has highlighted the influence of PE on the distribution of other lipid sub classes between leaflets by vectorial molecular probes [52]. By using the same approach, PE asymmetry and homeostasis under colistin treatment could be better characterized”.

7. All sections should be critically discussed and compared with the previous reports. This will actually strengthen the manuscript and will highlight the significance of the work.

In line with the remarks of reviewer 1’s suggestions and the answer to the question 6, the results were critically discussed and compared with previous results in the literature.

8. Conclusion is missing. Herein, I would like to see the major findings and how they are addressing the left behind research gaps and covering current challenges.

“Altogether, our present observations show that exposure of a clinical isolate of A. baumannii to sub lethal and lethal amounts of colistin triggers a set of lipid changes. Palmitic acid percentage increases only for susceptible strain suggesting an adaptation of the plasma membrane fluidity to limit colistin membrane destabilization. PE percentage decrease significantly on both strains, which, in turn provides pEtN for LOS modification. As described in the literature [45], this addition of pEtN moiety on lipid A reduces the overall net-negative charge of the bacterial surface and confers resistance to colistin. The present study provides an interesting view on lipid homeostasis of one clinical isolate (A. baumannii 721164) but the protocol can be extended to other clinical isolates. The 130 lipid species described could help future works dealing with A. baumannii lipidomics in general and on inner and outer membrane asymmetry in particular. Our efforts could contribute to the development of novel inhibitory molecules targeting bacterial lipids to fight the difficult-to-treat diseases associated with A. baumannii”

9. Literature needs to be updated with care.

The literature has been updated, just historical references remained.